# Effect of Hot-Alkali Treatment on the Structure Composition of Jute Fabrics and Mechanical Properties of Laminated Composites

**DOI:** 10.3390/ma12091386

**Published:** 2019-04-29

**Authors:** Xue Wang, Lulu Chang, Xiaolong Shi, Lihai Wang

**Affiliations:** College of Engineering and Technology, Northeast Forestry University, Harbin 150040, China; wangxue6025@gmail.com (X.W.); lu_romance@163.com (L.C.); Shi63260186@sina.com (X.S.)

**Keywords:** jute fabrics, hot-alkali treatment, laminated composites, mechanical properties, fracture surfaces

## Abstract

In this study, jute fabrics/epoxy-laminated composites were fabricated via a simple and effective manual layering. Hot-alkali treatment was used to pretreat jute fabrics to improve their interfacial compatibility. The effects of hot-alkali treatment with five concentrations (2%, 4%, 6%, 8% and 10%) on the composition, crystallinity and surface morphology of jute fibers, were analyzed with the aids of Fourier transform infrared spectroscopy (FTIR), X-ray diffractometry (XRD), and the scanning electron microscope (SEM). The mechanical properties (tensile and flexural) of laminated composites, and the morphology of the tensile fracture surface, were analyzed. The results indicated that the crystallinity index (CI) and crystallite size (CS) of the cellulose in jute fibers were improved, and there were three stages for CI and CS with the increase of alkali concentrations. Hot-alkali treatment improved the mechanical properties of laminated composites, especially for the 6% NaOH-treated jute fabric reinforced. The tensile strength, flexural strength, tensile modulus and flexural modulus of 6% NaOH-treated fabrics reinforced composites were enhanced by 37.5%, 72.3%, 23.2% and 72.2%, respectively, as compared with those of untreated fabrics reinforced composites. The fiber pull-out and the gaps of the tensile fracture surface were reduced after hot-alkali treatment.

## 1. Introduction

With the increasing depletion of petroleum resources, the development and utilization of new environment-friendly structural and functional materials has become imperative [1,2,3,4]. Nowadays, the environmental protection materials are advocated. The laminated composite is one of them. It is the superposition of fibers (carbon fibers, glass fibers [5] or plant fibers) and resin, by imitating the physiological structure of nacre (Mother of Pearl) layers. Some natural biological materials have excellent matching in structure and function, such as bamboo, wood, bones, natural fibers, nacre and others [6,7]. Especially, shell nacre is composed of organic substances and hydrated calcium salts (aragonite crystals) [8]. The aragonite crystals structures pile themselves up to form a very neat and orderly structure like a “brick wall” [9]. The nacre layers in a shell are natural laminated composites. Organic layers and aragonite crystals are the matrix and reinforcements of the natural laminated composites, respectively. Their combination gives a shell its high strength and toughness [10]. Scholars have enlightened the structure of organic-aragonite-organic in nacre to prepare the matrix-fabric-matrix laminated composites. 

Similar structures were found in beetle exoskeletons, which inspired scientists and engineers to develop the bionic design of laminated composite materials [11,12,13,14].

A kind of economical and environment-friendly reinforcement is the key to the preparation of laminated composite. It is worth noting that the volume and use of jute fibers are second only to cottons. The jute fiber cells are pentagonal or hexagonal in cross section, and have soundproofing, UV protection and antibacterial properties [15]. In addition, jute fabrics have good mechanical properties, and they are suitable as reinforcements for bionic, laminated composites. The jute fabric-reinforced composites meet the requirements for the structural properties of commercial materials at a relatively low cost [16].

However, the main components of jute fibers are cellulose, hemicellulose, lignin, pectin, and a few extracts [17,18]. The surfaces of natural fibers have a large number of hydrophilic groups, which need to be pretreated to improve the interfacial compatibility of composites. Commonly used surface treatment methods include heat treatment, alkali treatment, coupling agent treatment, and acetylation treatment [19]. Moreover, alkali treatment is a simple and effective way, because of its low cost, high efficiency, and convenience [20]. The tensile strength of composites increased by 65% via a 0.5% alkali-steam treatment [21]. In other works [22,23,24], the 5% alkali-treated jute fibers-reinforced epoxy composites had higher tensile strength than the 10% treated ones [22]. And after 5% alkali treatment for 2 h and 4 h, the flexural strength of composites were increased by 3.16% and 9.5%, respectively [23]. Due to the weakened hydrophilic behavior of jute fibers after alkaline treatment, the compatibility was accordingly improved [20]. Alkali treatment removed lignin and hemicellulose from natural fibers. In addition, the cellulose in the natural fiber was mercerized by the higher alkali solution to improve the crystal structure and stability. Mercerization caused the transformation of the crystal structure of cotton fibers from cellulose I to cellulose II [24]. The crystallinity of the 10% alkali-treated cotton fiber was slightly higher than that of the untreated cotton fiber, while the 15% and 20% alkali-treated cotton fibers had lower crystallinity. In addition, alkali treatment leads to an increase in the effective surface area between the fiber and the matrix, with the removal of impurities, hemicellulose and lignin on the surface of fibers [24,25,26]. 

In this study, the jute fabrics-reinforced laminated composites were fabricated via manual layering. The epoxy resin was used as matrix material. The hot-alkali treatment was used to improve the interface compatibility between jute fabrics and epoxy resins. The characterizations of hot-alkali-treated jute fabrics were analyzed by FTIR, SEM and XRD. The process and principle of hot-alkali treatment on jute fibers were also deeply discussed. The mechanical properties (tensile and flexural) of the jute fabric-reinforced laminated composites were investigated. And the tensile fracture surfaces of laminated composites were observed by SEM.

## 2. Materials and Methods

### 2.1. Materials

Epoxy resin E-44, polyamide resin 650, and thinner 692 were purchased from Zhenjiang Danbao Resin Limited (China) as the matrix of the composite. The plain jute fabrics supplied by Guangzhou Yin Fan Textile Limited (China) were used as continuous reinforcements. As shown in Figure 1, the jute fabric has the same warp and weft densities of 50/10 cm. 50 is the number of yarns per 10 cm in the warp and weft directions of jute fabric.

### 2.2. Surface Treatments

Under normal conditions, cellulose is relatively stable in a low-concentration alkali solution, but under high temperature conditions, cellulose is prone to alkaline degradation. Xia et al. reported [27] that 120 °C was the best alkali treatment temperature. In order to ensure that the lignin and hemicellulose of the jute fibers were effectively removed under the premise of not damaging the cellulose, the condition of the alkali treatment was set to 120 °C, for 90 min. The process of the surface treatment of jute fiber is as follows:

Jute fabrics (230 mm × 275 mm) were immersed in NaOH solutions with concentrations of 0%, 2%, 4%, 6%, 8% and 10% (by weight) at 120 °C for 90 min, respectively. Then the fabrics were rinsed with distilled water to scour off the residual lye until neutrality. After that, the jute fabrics were dried at 80 °C to constant weight. Figure 2A shows the principle of hot-alkali treatment. Alkaline solution removes most of the hemicellulose and lignin with unstable structure, and retains the cellulose with stable structure and better mechanical properties. Hot-alkali treatment also increases the crystallinity and effective surface area of the fibers.

### 2.3. Composite Fabrication

The surface-treated jute fabrics were prepared into laminated composites by a simple manual layering technique. Because the silicone cushion is resistant to high temperature, non-adhesive with resin, and easy to clean, so the silicone cushions were placed on the upper and lower surfaces of the laminated composite to assist the solidification of laminated composite. The preparation processes of laminated composites are shown in Figure 2B. Firstly, the upper and lower surfaces of two-layer jute fabrics were coated with mixed epoxy resin (Epoxy resin E-44: Polyamide resin 650: Thinner 692 = 10:6:1) on a layer of silicone cushion. Polyamide resin was the curing agent, and thinner 692 was used as the diluent. Then another silicone cushion was placed over the epoxy-coated jute fabrics and compacted to eliminate air entrapment. Finally, the composites with silicone cushions were put into a drying oven for curing at 100 °C for 90 min. Having removed the silicone cushions and cleaning, the fiber content of the laminated composite was 20% (in weight).

### 2.4. Characterizations

#### 2.4.1. Jute Fiber Composition Analysis

The percentage of cellulose, lignin, hemicelluloses, extractive and ash in the raw and treated jute fibers were evaluated by the following suitable techniques. The lignin, holocellulose and α-cellulose contents were determined by the Klason method (GB/T 2677.8-94, China), sodium chlorite method (GB/T 2677.10-1995, China) and pulps determination of alkali resistance (GB/T 744-2004, China), respectively. As for the other components, the content of hemicellulose was that of holocellulose minus α-cellulose. For the extractive and ash, they were evaluated by the Technical Association of the Pulp and Paper Industry (TAPPI).

#### 2.4.2. Fourier Transform Infrared (FTIR) Spectroscopy 

The chemical structures and compositions of treated and untreated jute fibers were analyzed by a Thermo Nicolet FTIR spectrometer. The jute fibers were mixed up with KBr and tableted. 

#### 2.4.3. X-ray Diffraction (XRD) Test

The crystallinity index (CI) and crystallite sizes (CS) of raw and treated jute fibers were analyzed by X-ray diffractometry (X’Pert Power, PANalytical, Netherlands) with a scan range between 10° to 90° (2θ angle). The CI and CS were calculated by following equations:(1)$$\mathrm{CI}=\left(1-\frac{I_{am}}{I_{002}}\right)\times 100\%$$
(2)$${\mathrm{CS}}_{002}=\frac{0.89\lambda }{\beta_{002}\cos\theta }$$
where *I_002_* is the intensity of crystalline phase peak at around 22°, *I_am_* is the intensity of the amorphous phase peak at around 16°, *λ* is the wavelength of the X-ray, *β_002_* is the full width half maxima, and *θ* is the angle of diffraction.

#### 2.4.4. Scanning Electron Microscope (SEM)

The surfaces of treated fibers and fracture surfaces of the laminated composites were observed by using a scanning electron microscope (SEM) model JSM 7500F. Before analysis, all samples were coated with a thin layer of gold to avoid samples arcing.

### 2.5. Physical Properties

#### 2.5.1. Density

Densities of laminated composites with alkali-treated and -untreated jute fabrics reinforced were calculated by:(3)$$\rho =\frac{m}{abc}\times 1000$$
where *ρ* (g﹒cm^−3^) is the density of the laminated composite, *m* (g) is the mass of the test specimen, and a, b, c (mm) are the length, breadth and thickness, respectively. The mass of the test specimen was weighed on an electronic balance at 20 °C and 65% RH. The length, breadth and thickness were measured by the Vernier caliper.

#### 2.5.2. Tensile Test

The tensile tests were conducted according to GB/T 1447-2005, using a universal testing machine (CMT5504). The specimens (3 mm × 25 mm × 250 mm) of each sample type were tested with a test speed of 5mm/min and the average values were reported. Figure 3 is the photo of these tensile test samples. Ten specimens of each sample were tested, and the average values were reported. The end of each sample was pasted with wooden reinforcing plates (2 mm × 25 mm × 50 mm). The tensile strength *σ_t_*, elongation at break *ε_t_* and tensile modules of elasticity *E_t_* were calculated by Equations (4)–(6), respectively.
(4)$$\sigma_{t}=\frac{F_{t}}{b\cdot c}$$
where *σ_t_* (MPa) is the tensile strength, *F_t_* (N) is the fracture load, b and c (mm) are the breadth and thickness of specimens of laminated composites.
(5)$$\varepsilon_{t}=\frac{△L_{b}}{L_{0}}\times 100\%$$
where *ε_t_* is the fracture strain, *L_0_* (100mm) is the gauge length of specimen, △*L_b_* (mm) is the tensile deformation of specimens at break within the scope of *L_0_*.
(6)$$E_{t}=\frac{L_{0}\cdot △F_{t}}{b\cdot c\cdot △L}$$
where *E_t_* (MPa) is the modulus of elasticity in tension, △*F_t_* (N) is the increment of load in the range of linear elasticity, *L_0_* (100 mm) is the gauge length of specimen and △*L* (mm) is the deformation increment corresponding to △*F_t_*.

#### 2.5.3. Flexural Test

The flexural strengths of laminated composites were measured using three point bending tests according to GB/T 1449-2005 by a universal testing machine (CMT5504). The specimens (3 mm × 25 mm × 125 mm) of each sample type were tested with a test speed of 2 mm/min and the average values were reported. Ten specimens of each sample were tested, and the average values were reported. The flexural strength *σ_f_* and flexural modulus *E_f_* were calculated by Equations (7) and (8), respectively.
(7)$$\sigma_{f}=\frac{3P\cdot l}{2b\cdot c^{2}}$$
where *σ_f_* (MPa) is the flexural strength, *P* (N) is the fracture load, and *l* (80 mm) is the span of flexural specimens.
(8)$$E_{f}=\frac{l^{3}\cdot △P}{4b\cdot c^{3}\cdot △S}$$
where *E_f_* (MPa) is the flexural modulus, △*P* (N) is the increment of flexural load in the range of linear elasticity, △*S* (mm) is the increment of deflection corresponding to △*P*.

## 3. Results and Discussion

### 3.1. FTIR Analysis of Raw and Treated Jute Fibers

The variances in chemical structures and compositions between raw and hot-alkali-treated (five incremental concentrations) jute fibers were qualitatively analyzed, and their different groups’ vibrations are shown in Figure 4. The peak at 3329 cm^−1^ is an O–H stretching vibration of both treated and untreated jute fibers [28]. The hydroxyl groups are present in the main components of the jute fiber (cellulose, hemicellulose, and lignin), and bond together in the form of intermolecular hydrogen bonds or intramolecular hydrogen bonds. The peak at 2918 cm^−1^ is a C–H stretching vibration of cellulose and hemicellulose [29], which becomes weaker after alkali treatment. It is clear that the treated jute fibers (HA2, HA3, HA4, HA5 and HA6) with hot-alkali have no obvious absorption peaks at 1735 cm^−1^ and 1235 cm^−1^. The reason is that absorption peak at 1735 cm^−1^ is the C=O stretching vibration of the carboxylic acid in hemicellulose of untreated jute fibers [28,30]. In addition, the band at 1235 cm^−1^ appears to be due to the C–O stretching vibration of the acetyl group in the lignin of untreated jute fibers. However most of the lignin and hemicellulose of these jute fibers were removed by hot-alkali treatment. The chemical comparison of raw and treated jute fabrics are shown in Table 1. The cellulose content of 6% NaOH-treated jute fabric improved from 53.69% to 75.52%. The content of hemicellulose and lignin decreased from 25.56% and 11.10% to 10.43% and 9.05%, respectively. And the hot-alkali treatment not only changed the chemical comparison of the jute fabric, but also increased the crystallinity, accompanied by the removal of hemicellulose and lignin.

### 3.2. XRD Analysis of Raw and Treated Jute Fibers

The X-ray diffraction patterns of jute fibers treated by five alkali concentrations at 120 °C are shown in Figure 5a. It exhibits that the diffraction peaks at around 15.4° and 22.4° belong to the (101) and (002) crystal faces of cellulose, respectively [28,29]. As a whole, it is also shown that the hot-alkali treatment did not change the crystal form of cellulose. But the diffraction peak shapes (002) of the crystallization region of cellulose become more sharp and narrow as the concentration of the alkali solution increased, and the diffraction peak shapes (101) of the amorphous region change inversely. There are two peaks at (101) for HA5 and HA6. This may be due to the changes of native cellulose I to cellulose II of jute fibers at higher NaOH concentrations (6%–10%), as shown in Figure 6f. The crystallinity index (CI) and crystallite size (CS) were calculated by Equations (1) and (2). There are three stages of change, as shown in Figure 5b. CI and CS of untreated jute fibers are 37.09% and 3.58 nm, respectively. They increase to 47.7% and 4.21 nm at 2% alkali treatment, but then decline to 45.77% and 3.92 nm at 4% alkali treatment, and increase again at higher concentrations (6%, 8%, 10%). 

Mwaikambo and Ansell also reported a similar observation [30]. Perhaps different concentrations of alkali solution are absorbed by different composite systems of jute fibers [31]. The fine structure of cellulose consists of crystalline regions and amorphous regions (Figure 6a). In the amorphous region, cellulose, lignin and hemicellulose are randomly arranged, and the layers of pectin, wax and impurities on the fiber surface are easily removed by a low-concentration alkali solution (Figure 6b) [28]. So the CI and CS of 2% NaOH-treated jute fibers are higher than untreated. But in the crystallization region, the cellulose chains are regularly arranged, and the spaces are filled with lignin and hemicellulose between adjacent cellulose chains, which makes it difficult for low-concentration alkali penetration [30]. Perhaps with the increase of alkali concentration, a part of lignin and hemicellulose in the crystalline region begins to degrade, forming gaps, and then the fiber bundles become dispersed (Figure 6c). Therefore, both the CI and CS of 4% NaOH-treated fibers are lower than that of 2% NaOH-treated fibers, but they increase again at 6%, 8% and 10% of NaOH (Figure 5b). Hot-alkali treatment affects the crystal structure and strength of fibers primarily through the mercerization of cellulosic fibers. The three stages of mercerization were swelling of microfibrils, destruction of the crystalline regions, and new crystalline lattices recombination [24]. Low concentrations of hot-alkali-removed (2%–4%) impurities, lignin and hemicellulose from the fiber surfaces. The high concentration of hot-alkali (6%–10%) mercerized with native cellulose, which recombined the cellulosic macromolecules (Figure 6f). Thereby, the crystallinities of cellulose were increased again with 6%–10% hot-alkali treatment. In addition, the removal of more lignin and hemicellulose between adjacent cellulose chains (Figure 6d,e), which reduced the distance between chains and formed hydrogen bonds to connect [23,32]. The SEM analysis (Figure 7) of jute fibers and tensile fracture surfaces (Figure 10) also prove that the fiber bundles were dispersed and thinner after hot-alkali treatment, which is helpful to improve the adhesion between fibers and epoxy.

### 3.3. SEM Analysis of Raw and Treated Jute Fibers

The surface morphologies of untreated and five concentrations alkali-treated jute fibers were analyzed by SEM. As shown in Figure 7a, the smooth surface morphologies of untreated jute fibers are covered with pectin, wax and impurities, which may reduce the contact area between jute fibers and resin [23,29,33]. However, there are many wrinkles on the surface of treated jute fibers Figure 7b–f. Figure 7g–i are the high magnification images of untreated and 6%, 10% hot-alkali-treated jute fibers. As shown in Figure 7h,i, there are many gaps and micro-voids on the treated jute fiber surface. These micro-voids and gaps provide access to the epoxy resins, which is beneficial for improving the interfacial compatibility and mechanical properties of laminated composites. As well as this, the surface of hot-alkali-treated jute fibers becomes clean and rough with the removal of pectin, wax and impurities. The reason is that hot-alkali treatment not only removed impurities, but also disrupted hydrogen bonds in the network structure [31] (Figure 6). The removal of hemicellulose and lignin from fiber cells released the internal constraints of fibers. The crystallinity was improved with the crystal structure reorganization of the cellulose [23,32]. Therefore, alkali treatment is beneficial to increase contact areas of fibers and improve the crystallinity and mechanical properties of composites.

### 3.4. Mechanical Properties of Laminated Composites

The mechanical properties (tensile and flexural) of laminated composites reinforced with untreated and five concentrations (2%, 4%, 6%, 8% and 10%) of hot-alkali-treated jute fabrics are shown in Table 2 and Figure 8. The standard deviation obtained is included in Table 2 and Figure 8. The result shows that the mechanical properties of the laminated composites increase first, and then decrease, and the fracture strain increases to 1.91 ± 0.18% from 1.55 ± 0.21% with the increasing concentration of the NaOH solution. Especially the mechanical properties of 6% NaOH-treated jute fabrics reinforced laminated composites are superior to others. The tensile strength, flexural strength, tensile modulus and flexural modulus of 6% NaOH-treated fabrics reinforced composites are enhanced by 37.5%, 72.3%, 23.2% and 72.2%, respectively, as compared with those of untreated fabrics reinforced composites. The reason is that hot-alkali treatment increases the contact area between fibers and epoxy resins and enhances the crystallinity of fibers with the impurities, lignin and hemicellulose, having been removed by alkali treatment at 120 °C.

It is worth noting that the mechanical properties of 8% and 10% NaOH-treated laminated composites are worse than 6% NaOH-treated ones. This is because that a high concentration of alkali solution can not only remove lignin and hemicellulose, but also change the native cellulose I to cellulose II [30,31]. Table 1 shows that the cellulose contents were changed from 53.69% to 75.52% and 51.82% with the 6% and 10% hot-alkali treatment, respectively. Another reason, Figure 7i shows that the 10% hot-alkali treatment causes excessive peeling of the surface of the fiber, which reduces the strength and toughness of fibers. The reaction of sodium hydroxide with hydroxyl group of cellulose is shown as the following Equation (9).
R-Cell-OH + NaOH → R-Cell-ONa + H_2_O(9)

Overall, the hot-alkali treatment improved the mechanical properties of laminated composites, and the bending properties are better than the tensile. This is because of the cross-woven structure of the jute fabric. Under the three-point bending (Figure 9a), the stress distribution at the section 1-1 of the laminated composite is shown in Figure 9b, and the normal stress σ and the shear stress *τ* are produced by the bending moment *M* and the shearing force *P/2*, respectively. The longitudinal fibers (Figure 9c) of jute fabrics can transmit and resist the shear stress *τ*. Therefore, the strength of the jute fabrics were fully developed.

### 3.5. SEM Analysis of Tensile Fracture Surfaces of Laminated Composites

Tensile fracture surfaces of raw and treated fabrics reinforced composites were observed by SEM analysis. There are many voids due to fiber pull-out and clear gaps between the fiber root and the epoxy on the fracture surface of the untreated sample, as shown in Figure 10a. Because there are a lot of hydroxyl and impurities on the untreated jute fabric surface, so it does not adhere well with epoxy. And the strength of jute fibers were not fully developed, under the action of tensile stress [34]. However, the hot-alkali treatment improved the interfacial compatibility between jute fabric and resin. Figure 10b–f show that the fiber pull-out and the gaps between fibers and epoxy are unnoticeable. Jute fibers are more dispersed in the resin than untreated fabrics reinforced composites. This is because the hot-alkali treatment removed pectin, wax and other impurities on jute fibers surfaces, and increased the surface roughness and effective surface areas of the fibers [35,36,37]. Also, the removal of lignin and hemicellulose, achieved by the treatment, improved the crystallinity of the cellulose. Therefore, hot-alkali treatment improved the adhesion between jute fibers and the matrix [38], and the tensile stress transferred well between them. These are consistent with the mechanical properties of laminated composites. The interfacial compatibility and mechanical properties of laminated composites were improved by hot-alkali treatment.

## 4. Conclusions

The jute fabrics reinforced epoxy laminated composites were successfully prepared by manual layering. Jute fabrics were treated with different hot-alkali concentrations to reduce their hydrophilic nature, and the analysis of fiber composition (Table 1), FTIR and XRD showed that the lignin and hemicellulose of the hot-alkali treated jute fibers were removed, and the crystallinities of treated fibers were improved. The content of cellulose was increased from 53.69% to 75.52% with the 6% hot-alkali treatment. Then it reduced to 51.82% with 10% hot-alkali treatment. Due to the mercerization, the cellulose changed from cellulose I to cellulose II in higher NaOH concentrations. Optimized recombination and mercerization of the crystalline structure of cellulose may be the reason, which resulted a new balance of hemicellulose, lignin and cellulose contents of the 10% hot-alkali treated, similar to untreated fibers. The surface morphologies of hot-alkali-treated jute fibers were analyzed by SEM. There were many wrinkles, gaps and micro-voids on the treated fiber surfaces, and the fiber become clean and rough with the removal of pectin, wax and impurities. The 10% hot-alkali treatment caused the excessive peeling of the surface of the fiber (Figure 7i).

The mechanical properties of the jute fabrics reinforced laminated composite treated with 6% NaOH are superior, as compared with other fabric-composites. This was because the proper concentration alkali solution removed the impurities, lignin and hemicellulose of jute fibers without degrading the cellulose. In addition, the laminated composite reinforced with plain jute fabric fully played its strength under the action of bending. The fracture surfaces of untreated and treated fabrics reinforced composites were observed by SEM analysis. The fiber pull-out and the gaps between fibers and epoxy of treated fabrics reinforced composites are unnoticeable, as compared with untreated fabrics reinforced composites. Then the interfacial compatibility of the laminated composite with hot-alkali treated was remarkably improved.

In summary, the hot-alkali treatment improved the mechanical properties of the composite from the following two points:
(a)Hot-alkali treatment removed the impurities, lignin and hemicellulose of the jute fibers and dispersed the fiber bundles to finer bundles, then increased the effective contact areas of the jute fibers.(b)The lignin and hemicellulose between adjacent cellulose chains were removed with a suitable concentration (6% to 8%) hot-alkali treatment. The crystalline structures of cellulose were improved by mercerization. Then the spacing of adjacent cellulose chains was shortened. Hydrogen bonds were formed to connect the adjacent cellulose chains. Thereby the crystallinity index (CI) and crystallite size (CS) of cellulose were increased, and so the strength of the fiber was improved.

## Figures and Tables

**Figure 1 materials-12-01386-f001:**
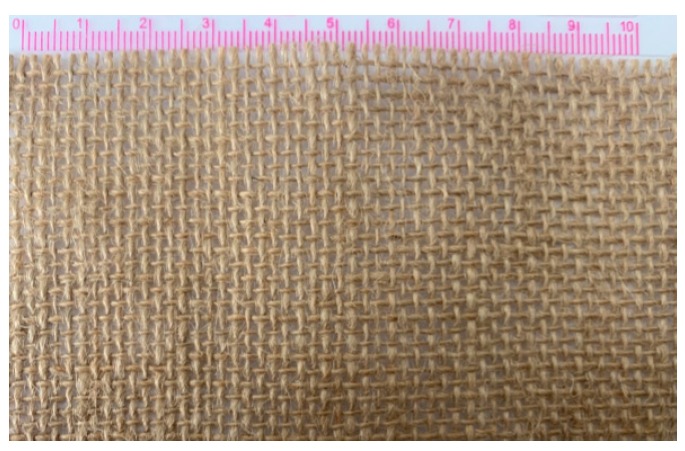
The plain jute fabric with the same density of warp and weft.

**Figure 2 materials-12-01386-f002:**
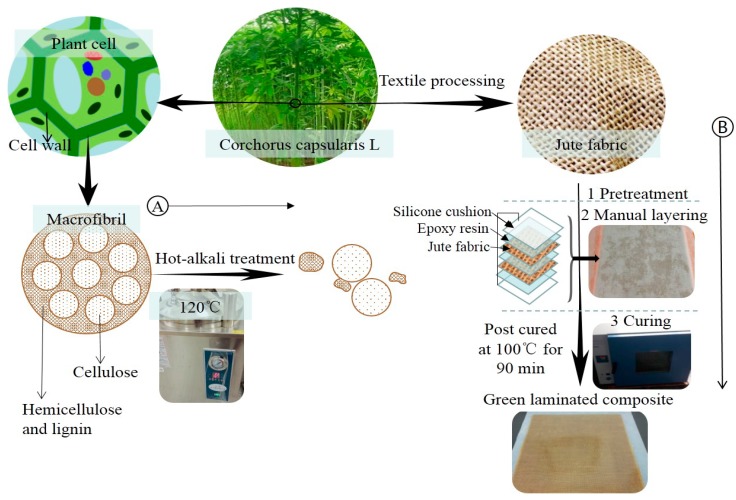
(**A**) is the principle of hot-alkali treatment, and (**B**) is the preparation processes of laminated composites.

**Figure 3 materials-12-01386-f003:**
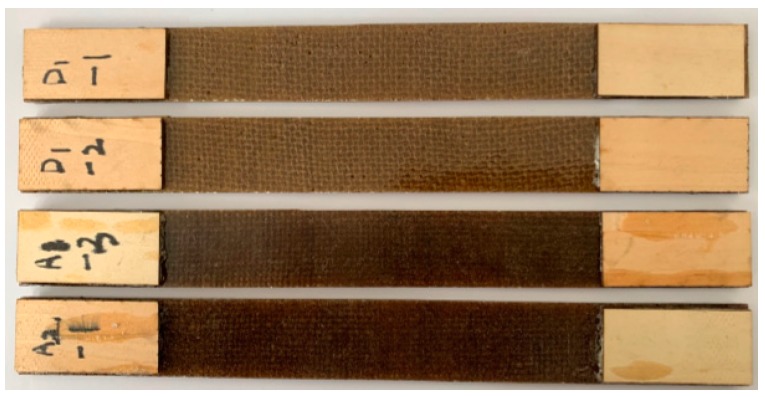
The photo of the tensile test samples, D1-1 and D1-2 are the untreated samples, A1-3 and A2-3 are the hot-alkali-treated samples.

**Figure 4 materials-12-01386-f004:**
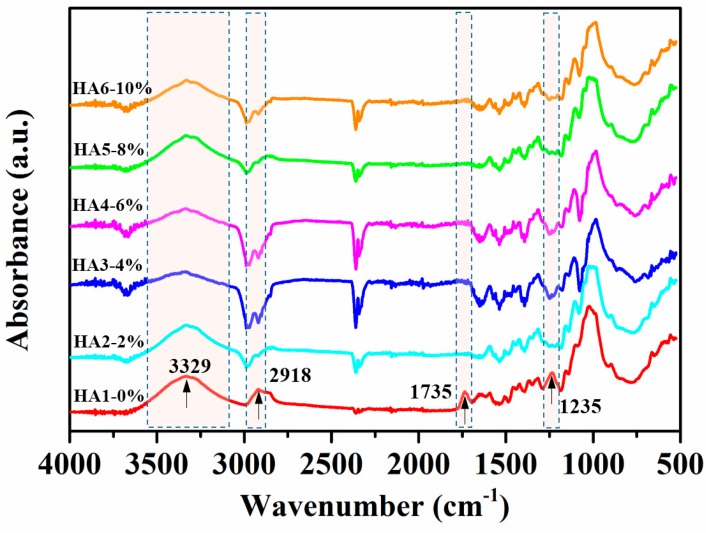
Fourier transform infrared spectroscopy (FTIR) spectra of raw (HA1-0%) and hot-alkali treated (HA2-2%, HA3-4%, HA4-6%, HA5-8%, HA6-10%) jute fibers.

**Figure 5 materials-12-01386-f005:**
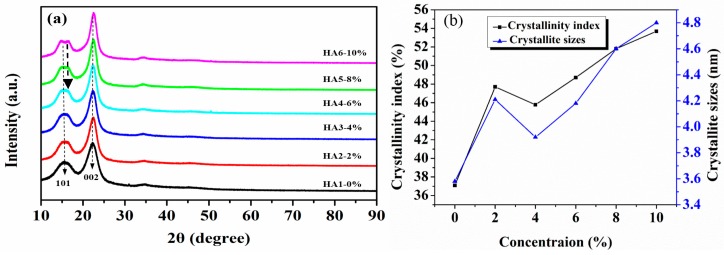
X-ray diffractometry (XRD) analysis of raw and hot-alkali treated jute fibers: (**a**) XRD spectrum, (**b**) crystallinity index (CI) and crystallite size (CS).

**Figure 6 materials-12-01386-f006:**
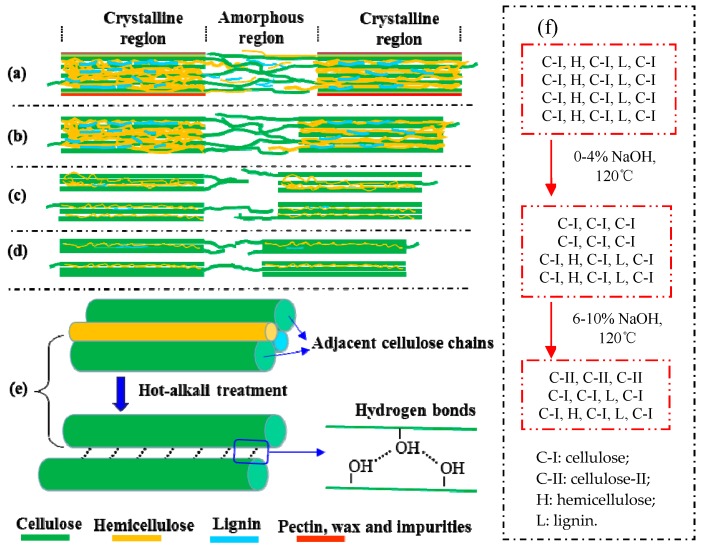
The schematic diagram of fine structure of cellulose and other polysaccharides of hot-alkali-treated jute fibers. (**a**) untreated, (**b**) 2% NaOH-treated, (**c**) 4% NaOH-treated, (**d**) 6%~10% NaOH-treated, (**e**) the adjacent cellulose chains, (f) the changes of cellulose, hemicellulose and lignin contents.

**Figure 7 materials-12-01386-f007:**
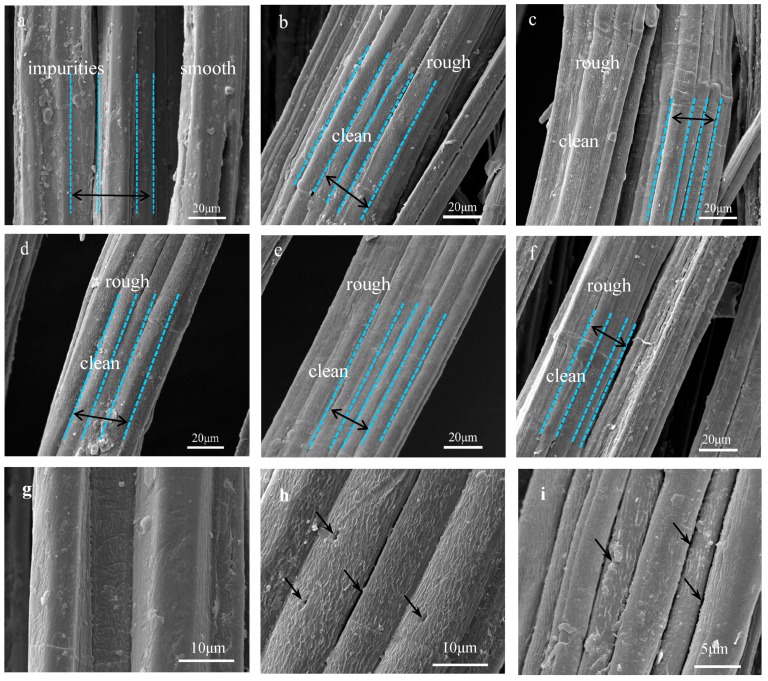
Scanning electron microscope (SEM) images of jute fibers surface: (**a**), (**g**) Untreated, (**b**) 2% NaOH-treated, (**c**) 4% NaOH-treated, (**d**), (**h**) 6% NaOH-treated, (**e**) 8% NaOH-treated, (**f**), (**i**) 10% NaOH-treated.

**Figure 8 materials-12-01386-f008:**
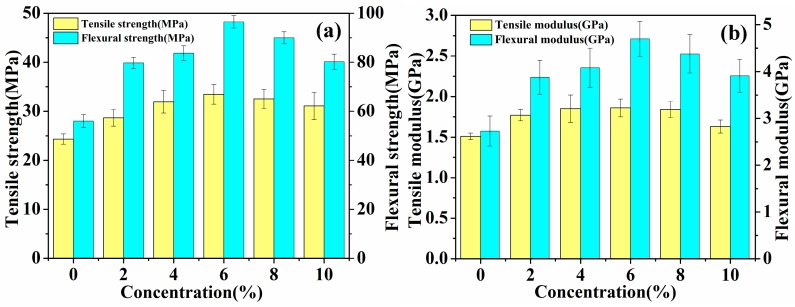
Mechanical analysis of laminated composites: (**a**) Tensile strength and flexural strength, (**b**) tensile modulus and flexural modulus.

**Figure 9 materials-12-01386-f009:**
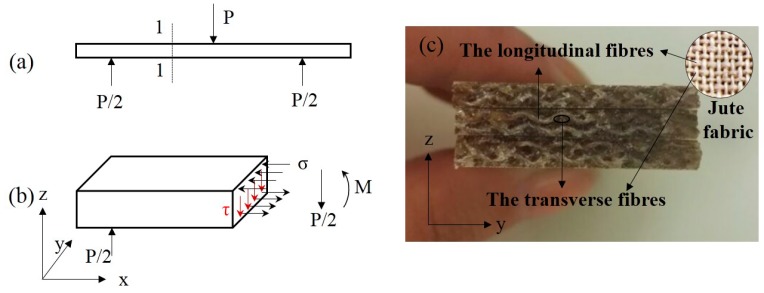
Schematic diagram of stress distribution at section 1-1 of the laminated composite under a three-point bending test. (**a**) Three-point bending test, (**b**) the stress distribution at section 1-1, (**c**) flexural fracture surfaces.

**Figure 10 materials-12-01386-f010:**
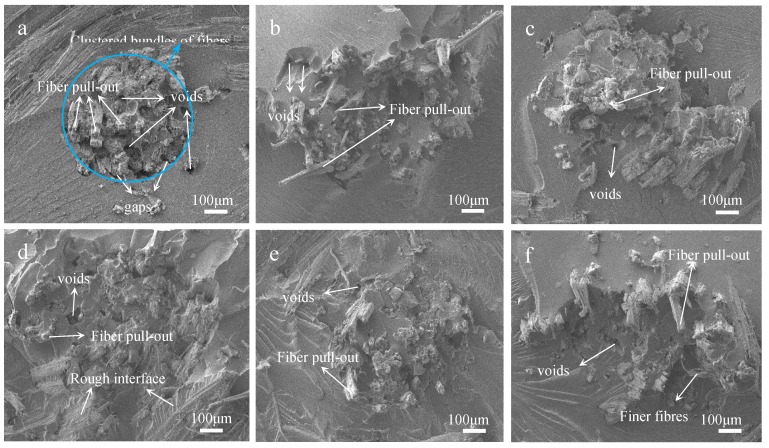
SEM images of fracture surfaces of laminated composites with hot-alkali treated jute fabrics reinforced: (**a**) 0% NaOH; (**b**) 2% NaOH; (**c)** 4% NaOH; (**d**) 6% NaOH; (**e**) 8% NaOH; (**f**) 10% NaOH.

**Table 1 materials-12-01386-t001:** Chemical comparison of raw and hot-alkali treated jute fabrics.

Fibers	Chemical Comparison (%)				
	Extractive	Cellulose	Lignin	Hemicellulose	Ash
untreated	3.89 (0.092 g)	53.69 (1.273 g)	11.10 (0.263 g)	25.56 (0.606 g)	0.67 (0.016 g)
2% NaOH	2.75 (0.064 g)	68.54 (1.589 g)	9.83 (0.228 g)	13.72 (0.318 g)	0.76 (0.018 g)
4% NaOH	2.67 (0.062 g)	73.40 (1.698 g)	9.12 (0.211 g)	11.07 (0.256 g)	0.76 (0.018 g)
6% NaOH	2.24 (0.052 g)	75.52 (1.744 g)	9.05 (0.209 g)	10.43 (0.241 g)	0.75 (0.017 g)
8% NaOH	2.66 (0.061 g)	65.96 (1.517 g)	8.97 (0.206 g)	16.62 (0.382 g)	0.75 (0.017 g)
10% NaOH	2.81 (0.062 g)	51.82 (1.145 g)	9.84 (0.217 g)	25.78 (0.570 g)	0.79 (0.017 g)

**Table 2 materials-12-01386-t002:** Mechanical properties of raw and alkali-treated jute fabrics reinforced composites.

Sample Number	Treatment of Fabric	Densities of Composites (g﹒cm^−3^)	Tensile Strength (MPa)	Tensile Modulus (GPa)	Fracture Strain (%)	Flexural Strength (MPa)	Flexural Modulus (GPa)
HA1	untreated	1.09 ± 0.01	24.34 ± 1.06	1.51 ± 0.04	1.55 ± 0.21	56.02 ± 2.63	2.73 ± 0.32
HA2	2% NaOH	1.10 ± 0.04	28.68 ± 1.70	1.77 ± 0.07	1.62 ± 0.12	79.73 ± 2.31	3.88 ± 0.36
HA3	4% NaOH	1.04 ± 0.01	31.96 ± 2.31	1.85 ± 0.17	1.73 ± 0.17	83.70 ± 3.01	4.08 ± 0.42
HA4	6% NaOH	1.08 ± 0.01	33.46 ± 2.01	1.86 ± 0.11	1.77 ± 0.15	96.50 ± 2.52	4.70 ± 0.37
HA5	8% NaOH	1.05 ± 0.02	32.51 ± 1.97	1.84 ± 0.10	1.80 ± 0.13	89.99 ± 2.45	4.38 ± 0.41
HA6	10% NaOH	1.11 ± 0.01	31.09 ± 2.77	1.63 ± 0.08	1.91 ± 0.18	80.24 ± 3.12	3.91 ± 0.35

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
