# Peer review of "Effect of Hot-Alkali Treatment on the Structure Composition of Jute Fabrics and Mechanical Properties of Laminated Composites"

_materials, 2019, doi:10.3390/ma12091386_

Round 1
Reviewer 1 Report
The paper is, in general terms, interesting, although not much originality and novelty is found. I' not sure if calling "green composite" to the obtained material is a right designation, as the matrix is still an epoxy resin.
There are some problems in the information (figure 1 really does not provide any useful information, as it is taken from bibliograhpy, from a material different to the one used in the study). Some language review should be performed (for example, lines 59/60, a "than" is missing; line 67 also needs some refining; lines 89-93... general language review); also it's hard to understand what is a "treated composite", and the authors mention some composites with PLA (although for others just "resin" is used); as neither the title of the paper neither the abstract introduces the type of polymer used, at this stage it is not clear wether this information is or not relevant. It is later stated than the chosen resin is an epoxy one, so some references about composites using this type of material should be included. There are many references that could be cited instead of the ones used in the paper (both for the fibre and the epoxy or the treatments).
What type of fabric is used: twill, plain? 1x2? 2x2?
How many test bars were used? Figure 3 does not provide any useful information.
For table 1, how do you explain the increased content in hemicellulose for 10 % treated fabric?
For figure 8/table 2, is there significant the mentioned difference for samples treated with 6 to 10% NaOH? Figures do not seem to be so different; also, the reduction in cellulose from 53.69 to 51.82 % is enough to lead to this difference?
Author Response
Dear reviewer:
Thank you for your comments concerning our manuscript entitled “Effect of hot-alkali treatment on the structure composition of jute fabrics and mechanical properties of laminated composites”. (ID: 459843). Those comments are all valuable and very helpful for revising and improving our paper, as well as the important guiding significance to our researches. We have studied comments carefully and have made correction, which we hope meet with approval. Revised portions are marked in red in the paper. The main corrections and the responds are as follows.
Thank you and best regards.
Yours sincerely,
Prof. Lihai Wang
College of Engineering and Technology, Northeast Forestry University.
Corresponding author: Lihai Wang Tel: +86 451 82190671
E-mail: wanglihai@nefu.edu.cn (L.W.); wangxue6025@gmail.com (X.W.).

Reviewer 2 Report
Authors describe some tests (chemical and mechanical) to find an optimal concentration of an alkaline treatment for jute fibers.
The overall paper is fine for publication in this journal but a few improvements can be made :
- on Figure 8, the scale for tensile values can be modified
- the link to nacre is not clear, especially looking at Figure 10
My last general concern would be with the process phase. It would have been very interesting to study the difference in impregnation of treated fibers. The amount of void or micro-void could explain differences in mechanical behaviours. I would expect to see that in further works.
Author Response
Dear reviewer,
Thank you very much for your constructive comments and suggestions to our manuscript entitled “Effect of hot-alkali treatment on the structure composition of jute fabrics and mechanical properties of laminated composites”. (ID: 459843). We are sure that your comments and suggestions are very important to improve the quality of our work. We have considered all your suggestions carefully. Revised portions are marked in red in the paper. The following is the detailed responses to comments.
Thank you and best regards.
Yours sincerely,
Prof. Lihai Wang
College of Engineering and Technology, Northeast Forestry University.
Corresponding author: Lihai Wang Tel: +86 451 82190671
E-mail: wanglihai@nefu.edu.cn (L.W.); wangxue6025@gmail.com (X.W.).

Reviewer 3 Report
The authors should mention at least the mercerization of cellulosic fibers by alkaline treatment (NaOH, ammonia), a process which affects both the morphology (crystal structure and crystallinity) and the physical characteristics (strength).
Cellulose I is the crystal structure of natural (untreated) cellulosic fiber and Cellulose II the crystal structure obtained after treatment with NaOH or after regeneration from solutions (e.g., of viscose or lyocell rayon fibers). Cellulose I is seen in Figure 5 for HA1-0% to HA3 -4%, while Cellulose II appears for higher NaOH concentrations, clearly in HA6-10%. The authors should discuss both crystallinity and related tensile properties based also on this morphological transformation.
As an example the authors are recommended to read and cite the following publication on mechanical properties of cellulose fiber-reinforced polyethylene oxide (PEO) composites showing that both original and mercerized cotton fibers enhanced the tensile strength of the PEO matrix:
YiYing You et al., Transition Properties of Cotton Fibers from Cellulose I to Cellulose II Structure, BioResources 8(4), 6460-6471 (2013).
Conclusions related to chemical composition of raw and hot-alkali treated jute fabrics should be drawn according to all data listed in Table 1. Why the % content of cellulose, hemicellulose and lignin of 10% NaOH treated jute fibers are practically very close to that of the original (untreated) fibers?
Author Response
Dear reviewer,
Thank you very much for your time on our manuscript entitled “Effect of hot-alkali treatment on the structure composition of jute fabrics and mechanical properties of laminated composites”. (ID: 459843). We appreciate your valuable comments and suggestions to our work. Those comments are all valuable and very helpful for revising and improving our paper, as well as the important guiding significance to our researches. We have considered all the comments and revised them point to point. Revised portions are marked in red in the paper. The detailed responses are as follows.
Thank you and best regards.
Yours sincerely,
Prof. Lihai Wang
College of Engineering and Technology, Northeast Forestry University.
Corresponding author: Lihai Wang Tel: +86 451 82190671
E-mail: wanglihai@nefu.edu.cn (L.W.); wangxue6025@gmail.com (X.W.).

Round 2
Reviewer 1 Report
Dear authors,
the manuscript quality has been increased after the review process; however, some minor changes still need to be performed:
line 137: a-cellulose should be changed to α-cellulose
New figures and text help understanding the paper and make it more consistent.
Reviewer 3 Report
Mentioning cellulose mercerization upon the treatment with NaOH and the change of the crystalline structures from I to II was a must. Just continue editing the text for minor corrections.